# Design of Energy Saving Controllers for Central Cooling Water Systems

Chang-Min Lee [1] , Tae-Youl Jeon [2] , Byung-Gun Jung [3] and Young-Chan Lee [4],*

1   Department of Marine System Engineering, Korea Maritime and Ocean University, 727, Taejong-ro, Yeongdo-gu, Busan 49112, Korea; oldbay@kmou.ac.kr
2   Korea Port Training Institute in Busan, Sinseon-ro 356-251, Nam-Gu, Busan 48562, Korea; terryjeon@gmail.com
3   Division of Marine System Engineering, Korea Maritime and Ocean University, 727, Taejong-ro, Yeongdo-gu, Busan 49112, Korea; bgjung@kmou.ac.kr
4   Department of Coast Guard Studies, Korea Maritime and Ocean University, 727, Taejong-ro, Yeongdo-gu, Busan 49112, Korea
*   Correspondence: yclee@kmou.ac.kr; Tel.: +82-410-4661

**Abstract:** Since fuel prices account for approximately 40% of a ship's operating costs, shipping companies worldwide have made significant efforts to save energy on board such as introducing new technologies or machine operation methods. Many ship operators have adopted an advanced control system using a variable-speed pump and/or an optimizing control system of a three-way valve on the outlet side of the central cooling system. It is often considered that the best way to control a central cooling system is to integrate the two control systems. However, when applied in practice, there is a frequent uncontrollable phenomenon in which the three-way valve is opened to its minimum and the variable-speed seawater pump is operated at its maximum, resulting in a large amount of energy consumption. Therefore, in this study, the speed of the variable-speed seawater pump is set to the minimum, and the feed-forward controller is adopted for the three-way valve control system. The input variable of the feed-forward controller is the Main Engine load, and it is designed to directly control the bypass openness with the three-way valve controller. Using this design, it was demonstrated that the variable-speed seawater pump was operated at a minimum and energy was saved.

**Keywords:** central cooling water system; heat exchanger; three-way valve; feed-forward control; variable-speed pump; energy saving on board ship

## 1. Introduction

### 1.1. Background

International oil prices have risen gradually since the first two oil price crises in the 1970s. Fuel prices have increased to approximately 40% of a ship's operating costs, thereby substantially impacting the shipping industry. Consequently, each shipping company has made various efforts to save energy. Such efforts include installing a seawater pump as a variable-speed pump in the new construction phase or by additionally installing a variable-speed control device to the existing seawater pump of an old ship [1–5]. Seawater supplied by the seawater pump is used majorly to recover waste heat from the cooling water of diesel ships. In this process, fresh water (cooling water) recovers waste heat from various heat sources in the ship and transfers it to seawater through a central cooling water cooler. Thereafter, it discharges the waste heat overboard [6,7]. To maintain a constant temperature of the cooling water supplied to the cooler of each part of the ship, waste heat is discharged by installing a three-way valve and a controller in the central cooling water cooler and by adjusting the opening of the three-way valve according to the cooling water temperature on the outlet side of the three-way valve [8]. A three-way valve is adopted as

a remedy of the temperature reversal in the water storage tank and a two-stage flowrate is implemented as the flowrate control strategy [9].

Likewise, another paper presents a model for a variable-speed vapor compression system that is able to accurately predict the minimal stable superheat (MSS) line. Based on bubble dynamics theory, a critical stable condition is defined, which the critical operating parameters of the evaporator should satisfy. Together with the model of the refrigeration system, the MSSs, depending on different operating conditions, are predicted successfully. The proposed model has been validated experimentally with steady state tests, presenting a prediction error lower than 5%. It is limited to just using MSS, and it could not control the variable speed input [10]. Other research presents an account of an investigation into the performance of multi-loop interacting digital self-tuning digital controllers for electromechanical drives incorporating a DC motor. It just uses its results of the plant and does not use other direct loads. It has the disadvantage of delivering the time to correct actual value to the set value [11].

A data-driven process controller is designed and implemented onboard a typical marine vessel for optimal variable-speed pump operation, leading to the energy efficiency optimization of its central cooling water system. To match variable flow rate requirements due to changes in the vessel's operational profile with respect to plant limitations, real-time process measurements are used as feedback signals to adjust the parameters and set-points of a data-driven process controller with self-tuned proportional-integral-differential control loops. It used actual results as a new set value for self-tuning, and programmed input data are also given to the plant. However, there is a limitation to overcome sudden M/E and G/E load fluctuation [12].

The variable speed seawater pump saves pump driving energy by controlling the frequency and voltage of the power supplied to the seawater pump driving motor according to the cooling water temperature at the outlet side of the three-way valve [13].

Proportional integral derivative (PID) controllers are usually used as variable speed seawater pump controllers and as a three-way valve control system. However, using the same control process variable, which is the cooling water outlet temperature, interference occurs between the two control systems [14], i.e., the three-way valve control system and the variable speed seawater pump control system. This interference makes it difficult to achieve the original purpose of energy saving.

In fact, in the training ship "Hannara" selected as the model for this study, there is a problem that the variable-speed seawater pump operated at 100% load, and the three-way valve is slightly opened to the cooler side because of interference between these controllers.

In this study, to reduce the flow rate of the variable-speed seawater pump and save energy, a feed-forward controller responding to disturbance fluctuations from the main engine load to the three-way valve control system was installed to initially control the three-way valve.

### 1.2. Study Content

The experiment was conducted using Mathwork's Simulink based on the voyage data and set values of the Hannara. Figure 1 demonstrates a schematic of the central cooling water system used in this study.

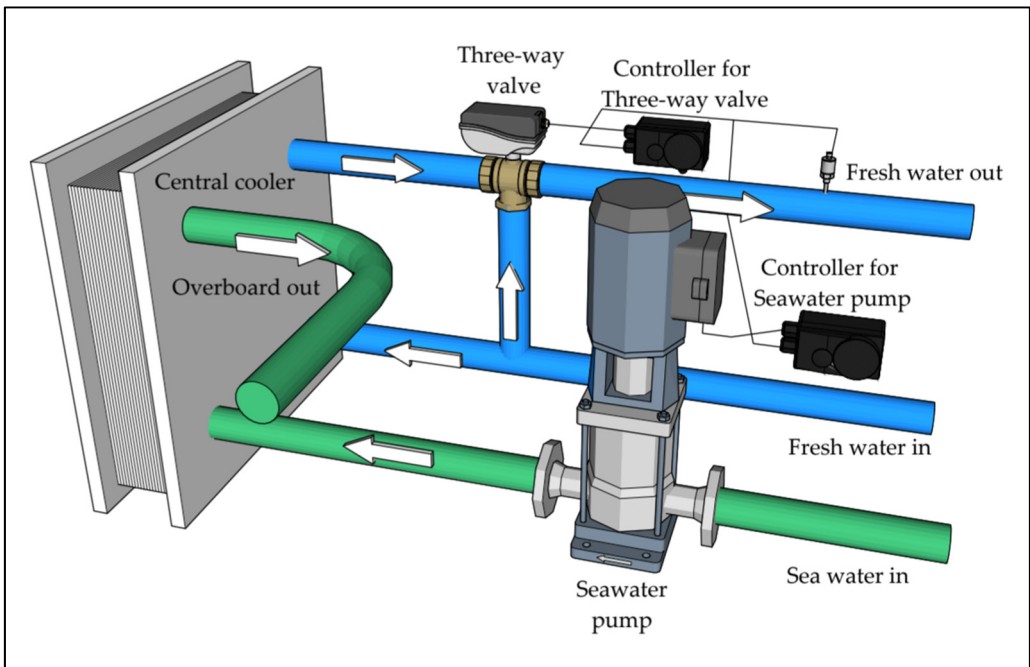

**Figure 1.** Configuration of the ship's central cooling system.

For the modeling of the central cooling water system, related data based on the cooling water system of the training ship Hannara were used. The configuration of the system was based on the thermal conductivity, heat transfer area, and heat distribution design drawings provided by the manufacturer, and the transfer function of the configuration device was simplified for the first delay system.

The experiment was conducted using Mathwork's Simulink based on the voyage data and set values of the Hannara.

The remainder of this paper is organized as follows: Section 2 describes the central cooling water system. Section 3 presents the basic theory and composition for modeling the central cooling system. The experiment and results are described in Section 4. Finally, a conclusion is drawn in Section 5.

## 2. Central Cooling Water System

In the past, ships used seawater to directly recover waste heat generated during operation. However, the method of direct cooling with seawater has disadvantages, such as reducing the efficiency of the cooler due to contamination of the heat transfer surface and of the pores due to continuous corrosion [15].

Owing to the development of the cooler cooling efficiency, most diesel ships in recent years have adopted a method of recovering the ship's waste heat using chemically treated cooling water and antifreeze, and then discharging the recovered heat into seawater through a central cooling water cooler [16,17].

The heat exchanger to be covered in this paper will utilize a plate-type cooler that cools the temperature of the ship's cooling water using seawater [18–21]. Figure 2 demonstrates the configuration of a plate heat exchanger that performs heat exchange between fluid and fluid and a temperature controller to maintain the cooling water temperature.

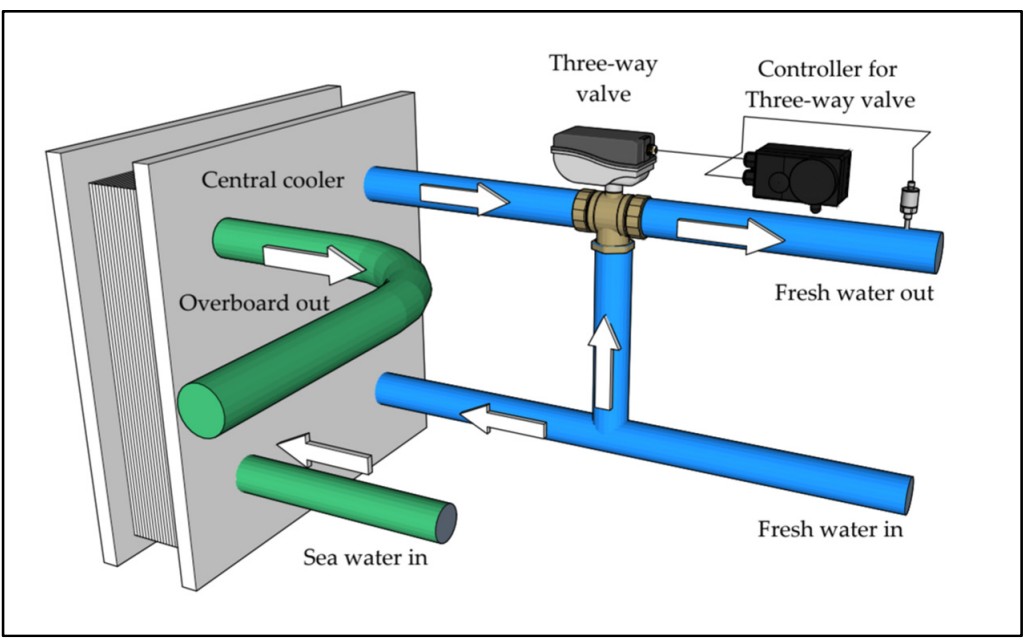

**Figure 2.** Configuration of central cooling water system.

## 3. System Modeling

In ships, waste heat is generated from the main engines, generators, and various auxiliary machinery. To discharge such waste heat to the outside of the system, a central temperature cooling water system is used. In the shipbuilding stage, the heat load generated by the ship is calculated in advance, followed by the central cooler capacity, cooling water pump capacity, and seawater pump capacity. Accordingly, a piping system is constructed.

Figure 3 demonstrates an example of the heat balance of a model ship.

Here, Q, H, and T denote the flow rate, quantity of heat, and temperature, respectively. The heat load is depicted considering that the main engine is operated under full load with two generators being operated.

The remarkable loads are "M/E load, M/E H.T J.W. cooler", and "M/E charge air Cooler" as the main engine load factors, and "L.O cooler", "CYL jacket", and "Air cooler" as the generator load factors [22].

Figure 4 categorizes the central cooling system, constructed into heat load, heat exchanger, three-way valve, and seawater pump.

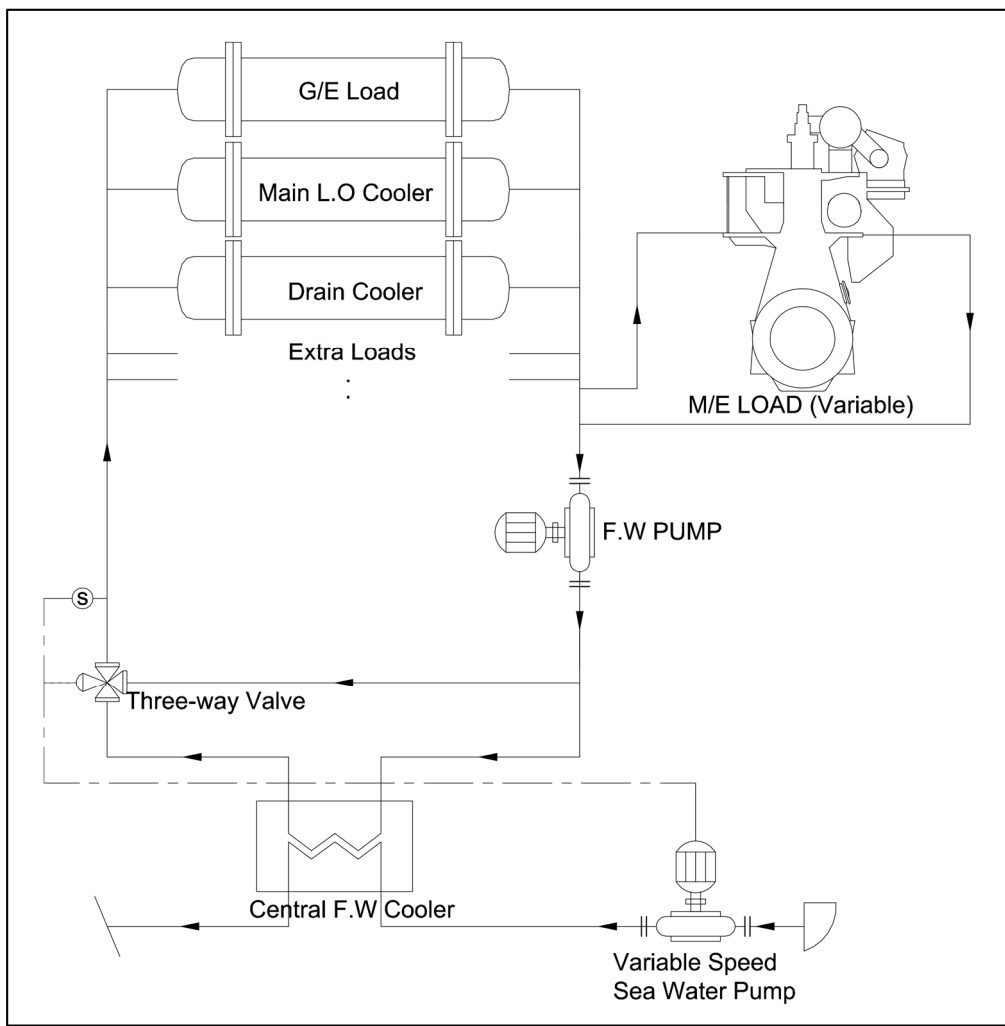

**Figure 3.** Heat balance for model ship.

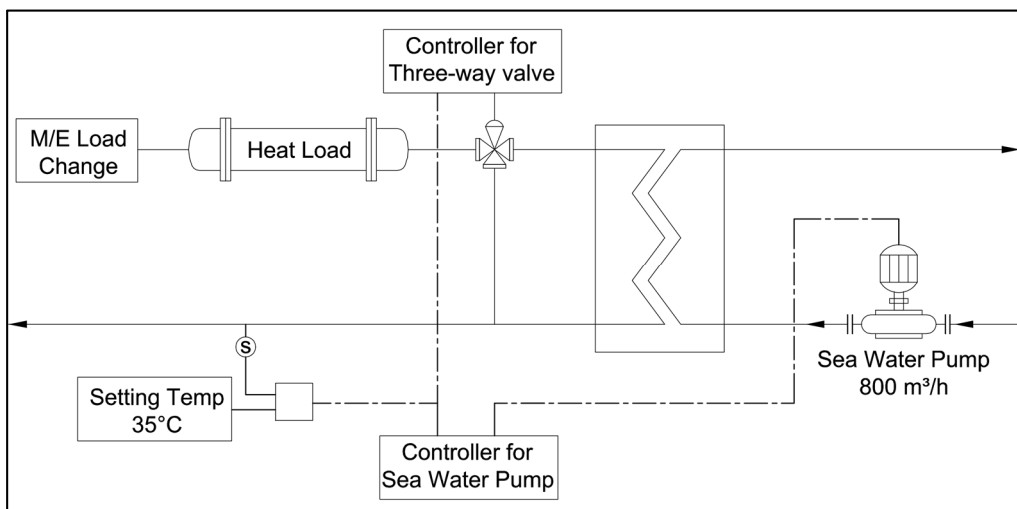

**Figure 4.** Model of central cooling system.

First, the cooling water that has recovered waste heat from the heat load enters the inlet of the three-way valve. In the three-way valve system, part of the cooling water enters the central cooling water cooler, whereas the rest is bypassed, and merges with the cooling water from the central cooling water cooler, after which it flows back to the heat load side.

The opening degree of the three-way valve is determined by the control value of Controller for Three-way valve, which takes the error between the three-way valve outlet temperature and the desired value (Setting Temp 35 °C) as input.

As for seawater, the flow rate increases or decreases as the power frequency of the seawater pump is controlled by Controller for Seawater pump, according to the same error used in the three-way valve.

Seawater discharged from the seawater pump enters the seawater side of the central cooling water cooler, recovers waste heat from the central cooling water, and discharges it overboard [22,23].

### 3.1. Heat Load

Here, we look at the input/output configuration of the heat load entering the central cooling system. Figure 5 shows the heat load of the model ship mentioned in Figures 3 and 4.

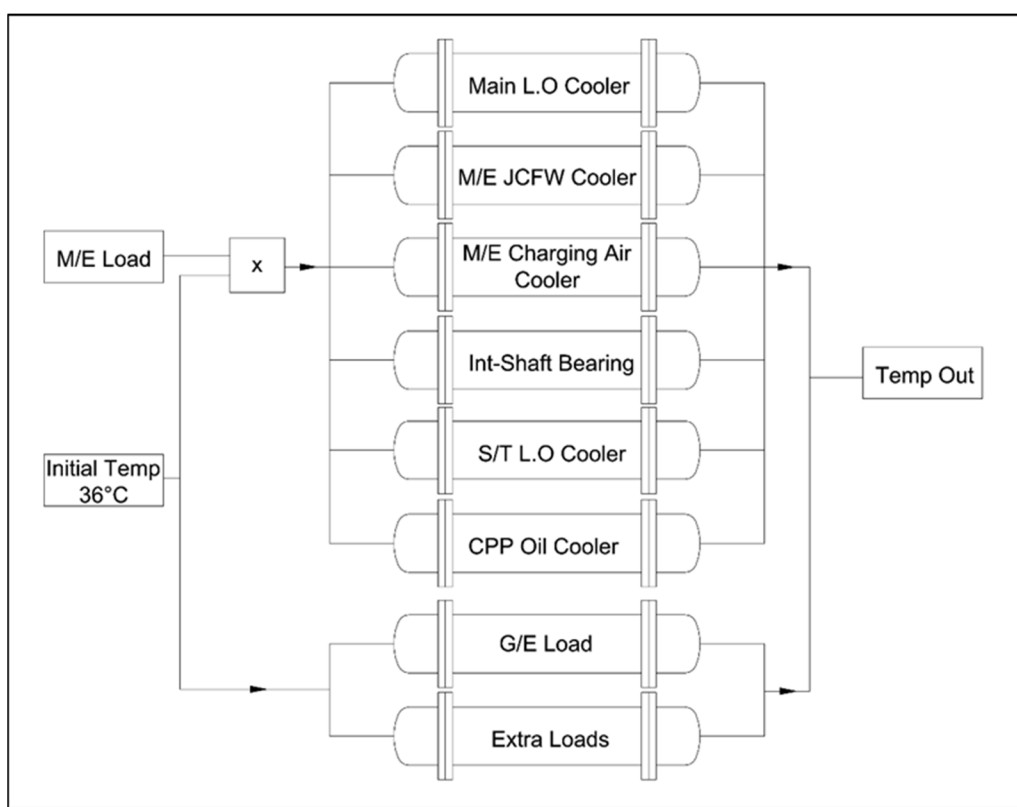

**Figure 5.** Heat load in central cooling system.

In the figure, the inputs are the initial temperature (Initial Temp) and load change (ME Load), and the output is the cooling water temperature (Temp out). Here, the density and specific heat are assumed to be the same.

Density, specific heat, and initial temperature are input as constant values, whereas the load change is a variable input, as illustrated in Figure 6. The data related to the load change are extracted from the operation information at the time of arrival and departure of the actual model ship.

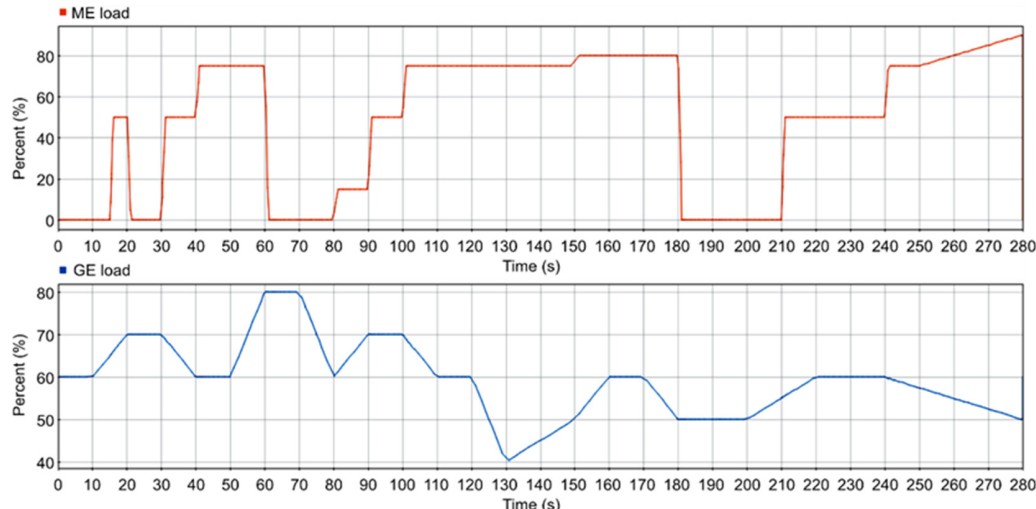

**Figure 6.** Variable input of M/E and G/E load.

The load change (ME Load) is the input to the load side of the "M/E LO Cooler," "M/E Charging Air Cooler," "Int-Shaft Bearing," "S/T LO Cooler," "CPP Oil Cooler," and "MGO Cooler for M/E SCR Blower Unit," whereas the temperature is the output from each heat exchanger.

In addition, the load related to the generator is configured, as demonstrated in Figure 5, inside the D/G Cooler. In the figure, the generator load change (Generator Load) is the input to the load side of the "L.O COOLER," "CYL JACKET," and "AIR COOLER," whereas the temperature is the output from each heat exchanger.

All other loads are configured as Etc (Fixed Load), as demonstrated in Figure 5, and output at a constant temperature.

Each output temperature (T) is divided by each output flow rate (Flow) and heat capacity ($m_c$) multiplied by specific heat and density; the final output is the composite temperature ($T_m$). The details are covered in Section 3.3. The output of the flow rate is obtained by summing all the values provided in the specifications of each heat exchanger.

Figure 6 is a scaled-down drawing of the section in which the ship uses the main engine load during one voyage. In general, it takes 1.5 to 2 h when the ship departs and sails in the navigation full ahead (R/up Eng'), but in this paper, is operating by. This simulation experiment was carried out under the assumption that rapid control of the opening of the three-way valve and the variable speed of the pump is performed in response to sudden load fluctuations of the main engine during a short period of time (280s). it is also differently controlled in the normal voyage of the ship.

For example, the operation time of the main engine and the generator load was set to 280 s, but it does not operate as such in the actual operation of a ship. The duration of departure is from 0 to 100 s, and from 100 to 180 s is the R/up Eng' section of normal voyage. In addition, at 180 s, the load of the main engine was suddenly dropped, and the load section was set to prove the effectiveness of the feed forward PID controller proposed in this paper. In addition, this was set up to re-verify the effectiveness of the proposed controller by increasing the load step by step from 180 s to 240 s.

### 3.2. Heat Exchanger

Here, we examine the input/output configuration of the central cooler, one of the elements depicted in Figure 4

The heat exchanger can be expressed, as shown in Figure 7, assuming that thermal equilibrium is always achieved in the heat exchanger according to the first law of thermodynamics.

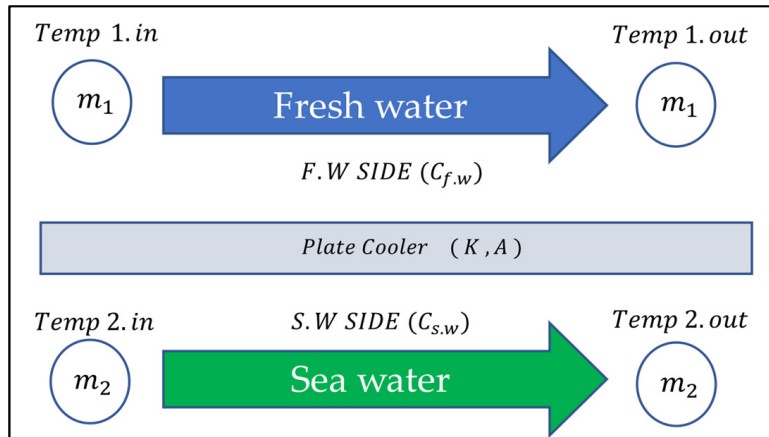

**Figure 7.** Heat flow of plate type cooler.

As the fluids $m_1$ and $m_2$ flow from the inlet (in) to the outlet (out), respectively, heat exchange is performed through the heat transfer plate.

Here, the heat energy released when the fluid $m_1$ moves from the inlet to the outlet and the heat energy recovered when the fluid $m_2$ moves from the inlet to the outlet are the same as the heat energy transferred from the fluids $m_1$ to $m_2$.

Therefore, the heat energy transferred from fluid to fluid can be expressed as Equation (1) [24,25].

$$Q = \frac{R_0 R_c}{R_0 + R_c} \times \Delta(Temp_{1.in} - Temp_{2.in}) \tag{1}$$

The smaller of the heat capacities $R_1$ and $R_2$ of the two fluids flowing in the cooler, $R_0$ is the product of the flow rate $m_0$ and the specific heat $C_w$.

$$R_0 = m_0 C_w \tag{2}$$

The transfer capacity of the cooler plate $R_c$ is multiplied by the heat transfer rate $K$ and area $A$.

$$R_c = KA \tag{3}$$

Substituting this into Equation (1), the heat energy $Q$ can be obtained as Equation (4) by multiplying the parallel sum of the heat capacity of the fluid $R_0$ and the transfer capacity of the cooler plate $R_c$ with the difference in inlet temperature of each fluid.

$$Q = \frac{m_0 C_w KA}{m_0 C_w + KA} \times \Delta(Temp_{1.in} - Temp_{2.in}) \tag{4}$$

The temperature changes $\Delta T_1$ and $\Delta T_2$ from the inlet to the outlet of each fluid can be obtained by dividing the heat capacities $R_1$ and $R_2$ of each fluid by the total heat energy $Q$ transferred per hour.

$$\Delta T_1 = \Delta(Temp_{1.in} - Temp_{1.out}) = \frac{Q}{R_1} \tag{5}$$

$$\Delta T_2 = \Delta(Temp_{2.in} - Temp_{2.out}) = \frac{Q}{R_2} \tag{6}$$

By subtracting the temperature change $\Delta T_1$ from the inlet temperature $Temp_{1.in}$ and adding $\Delta T_2$ to the inlet temperature $Temp_{2.in}$, the temperatures $Temp_{1.out}$ and $Temp_{2.out}$ of each fluid are obtained as the output.

$$Temp_{1.out} = Temp_{1.in} - \Delta T_1 \tag{7}$$

$$Temp_{2.out} = Temp_{2.in} + \Delta T_2 \tag{8}$$

In the central cooling water cooling system, $R_1$ is the heat capacity of cooling water and $R_2$ is the heat capacity of seawater. As input values, seawater has a temperature of 27 °C and a specific heat of 0.94 kcal/kg °C. The heat transfer rate, heat transfer area of the central cooler, and the number of coolers used are 6092 kcal/m²h °C, 90.4 m², and two pairs, respectively [26].

The heat transfer rate is 6092 kcal/m²h °C (7167 × 0.85), assuming a heat transfer rate of 7167 kcal/m²h °C and 85% of the dirt factor of the cooler. The calculated values are the output to the outside of the central cooler as $T_{f.w\ out}$ (Fresh Water Temperature Out) and $T_{s.w\ out}$ (Sea Water Temperature Out).

Substituting the values given in Equation (4), the amount of heat is:

$$Q = \frac{m_{sw} \times C_{sw} \times K_c \times A_c \times \Delta\left(Temp_{sw.in} - Temp_{fw.in}\right)}{m_{sw} \times C_{sw} + K_c \times A_c}$$

Substitute the given values, and weight the number of coolers and 85% efficiency.

$$Q = \frac{m_{sw} \times 0.94 \times 7167 \times 90.4 \times 2 \times 0.85 \times \Delta\left(27 - Temp_{fw.in}\right)}{m_{sw} \times 0.94 + 7167 \times 90.4 \times 2 \times 0.85}$$

Assume $K_m$ as;

$$K_m = 0.94 \times 7167 \times 90.4 \times 2 \times 0.85 \cong 1035339$$

$$Q = \frac{m_{sw} \times K_m \times \Delta\left(27 - Temp_{fw.in}\right)}{m_{sw} \times 0.94 + \frac{K_m}{0.94}}$$

Dividing the denominator and numerator by 0.94 is as follows.

$$Q = \frac{m_{sw} \times 0.94 \times K_m \times \Delta\left(27 - Temp_{fw.in}\right)}{m_{sw} \times 0.94^2 + K_m}$$

$$= \frac{m_{sw} \times 0.94 \times 1035339 \times \Delta\left(27 - Temp_{fw.in}\right)}{m_{sw} \times 0.940^2 + 1035339}$$

Using Equations (5) and (7) the fresh water outlet temperature is as follows.

$$T_{f.w\ out} = Temp_{fw.in} - \frac{m_{sw} \times 0.94 \times 1035339 \times \Delta\left(27 - Temp_{fw.in}\right)}{m_{sw} \times 0.940^2 + 1035339}$$

$$\times \frac{1}{m_{fw} \times 1.0}$$

$$= Temp_{fw.in} - \frac{m_{sw} \times 973218.66 \times \Delta\left(27 - Temp_{fw.in}\right)}{m_{fw}\left(m_{sw} \times 0.940^2 + 1035339\right)}$$

Using Equations (6) and (8) the seawater outlet temperature is as follows.

$$T_{s.w\ out} = Temp_{sw.in} + \frac{m_{sw} \times 0.94 \times 1035339 \times \Delta\left(27 - Temp_{fw.in}\right)}{m_{sw} \times 0.940^2 + 1035339}$$

$$\times \frac{1}{m_{sw} \times 0.94}$$

$$= Temp_{sw.in} + \frac{1035339 \times \Delta\left(27 - Temp_{fw.in}\right)}{m_{sw} \times 0.940^2 + 1035339}$$

### 3.3. Three-Way Valve

In Figure 8, the bypass amount of the three-way valve is increased or decreased by adjusting the opening degree with the controller according to the cooling water outlet temperature of the three-way valve.

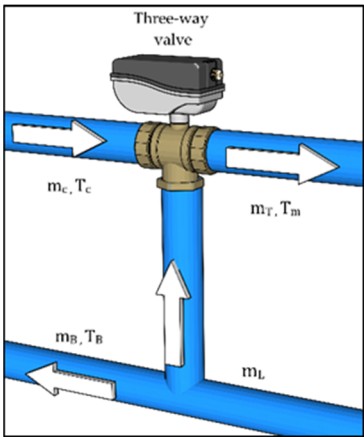

**Figure 8.** Cooling water flow in three-way valve.

Therefore, the difference between the bypassed flow rate ($m_B$) and the flow rate coming from the load ($m_L$) is the flow rate of heat exchanged in the cooler ($m_c$) [27].

$$m_L = m_T = m_B + m_C \tag{9}$$

If the temperatures of the fluid ($m_c$) from the outlet of the cooling water cooler and the bypassed fluid ($m_B$) are $T_c$ and $T_B$, respectively, the temperature of the two fluids mixed in the three-way valve is expressed as Equation (10) according to the heat transfer effectiveness and number of transfer units (NTU) method.

$$T_m = \frac{m_B c_B T_B + m_c c_c T_c}{m_B c_B + m_c c_c} \tag{10}$$

%B represents the degree of opening to the bypass side. Using %B, Equation (10) can be expressed as:

$$T_m = \frac{\%B m_L c_B T_B + (1-\%B) m_L c_c T_c}{\%B m_L c_B + (1-\%B) m_L c_c} = \frac{\%B T_B + (1-\%B) T_c}{\%B + (1-\%B)}$$

$$= \frac{\%B T_B - \%B T_c + T_c}{\%B - \%B + 1} = \%B (T_B - T_c) + T_c$$

## 4. System Experiment

### 4.1. System Composition

First, to observe the problems occurring in the existing central cooling water cooling system, an experiment is conducted based on the configuration shown in Figure 8. The cooling water outlet set temperature is set to 35 °C, and the temperature of seawater supplied to the variable-speed seawater pump is set to 27 °C.

The fluctuation of the heat load entering $m_L$ in Equation (9) is based on the actual operation data for the load fluctuation of the main engine. Figure 6 shows the load of the main engine over time, which is used as the input of the heat load fluctuation.

### 4.2. System Composition Results

First, the parameters of the controller are tuned to obtain a value similar to that of the model ship. The main engine load fluctuation depicted in Figure 6 was used as an input, and an experiment was performed through Simulink to obtain the result demonstrated in Figure 9.

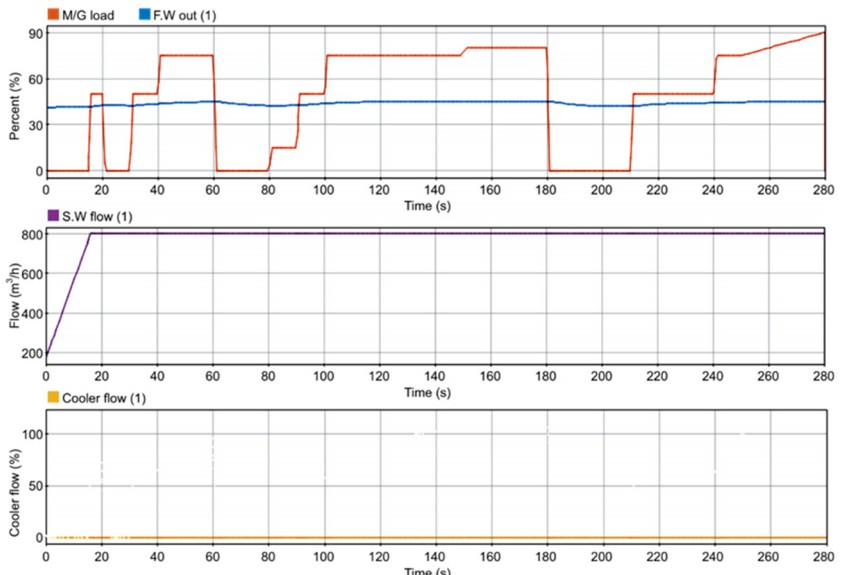

**Figure 9.** Actual condition of central cooling system.

The first graph in the figure shows the change in the load fluctuation input (M/E LOAD) of the main engine over time and the change in FW out (1), which is the cooling water temperature at the outlet of the three-way valve.

The second graph shows the SW flow (1), which is the change in flow rate of the seawater pump over time.

The third graph shows the Cooler flow (1), which is the opening degree of the three-way valve on the cooler side over time.

As with the model ship, it can be seen that the flow rate of the seawater pump is operated with the flow rate fixed at 100% due to the interference between the variable-speed seawater pump controller and the three-way valve controller. The opening of the three-way valve is controlled such that it is slightly opened towards the cooler.

In addition, the cooling water outlet temperature does not normally control the desired value (35 °C), and errors continue to occur.

Figure 9 is the flow rate of the variable speed seawater pump that appears in the model ship Hannara when the PID controller is installed according to the M/E and G/E loads described in Figure 6, and the bottom graph shows the three-way opening. To explain again, the graph at the top of Figure 9 shows that the temperature of F.W (blue line) is constant even with the load fluctuations of M/E and G/E, so that control is more or less performed. However, the middle graph shows the total amount of the flow rate of the variable speed seawater pump, and it can be seen that the variable seawater speed pump consumes a lot of energy by operating full load. In addition, it can be seen that the cooler three-way valve is operated in an almost closed state so that proper control is not achieved.

*4.3. System Composition with Feed-Forward Controller*

To solve the problems mentioned in the previous section, we intend to improve the control performance by adding a feed-forward controller to the existing system.

Figure 10 shows the configuration of the system in which the feed-forward controller using the main engine load as the disturbance process variable is combined with the existing system in Figure 4.

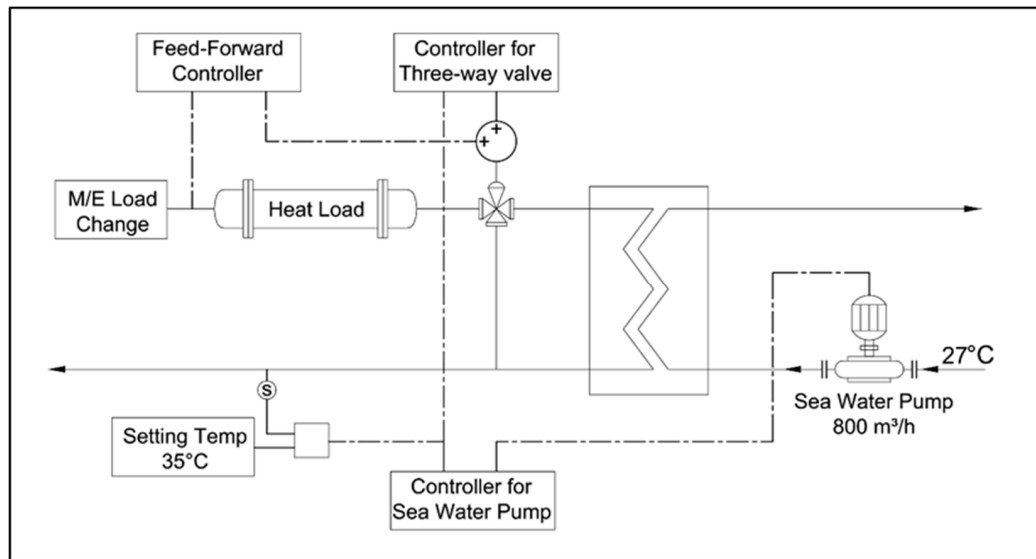

**Figure 10.** Central cooling system with feed-forward controller.

This is the addition of a feed-forward controller that uses the output signal from the "ME Load change" as an input value to the system configured, as demonstrated in Figure 6. The output of the feed-forward controller and the output of the three-way valve controller are added to form a system that serves as the input of the three-way valve.

### 4.4. System Composition Results with Feed-Forward Controller

The experimental results for the system in Figure 10 combined with the feed-forward controller are shown in Figure 11.

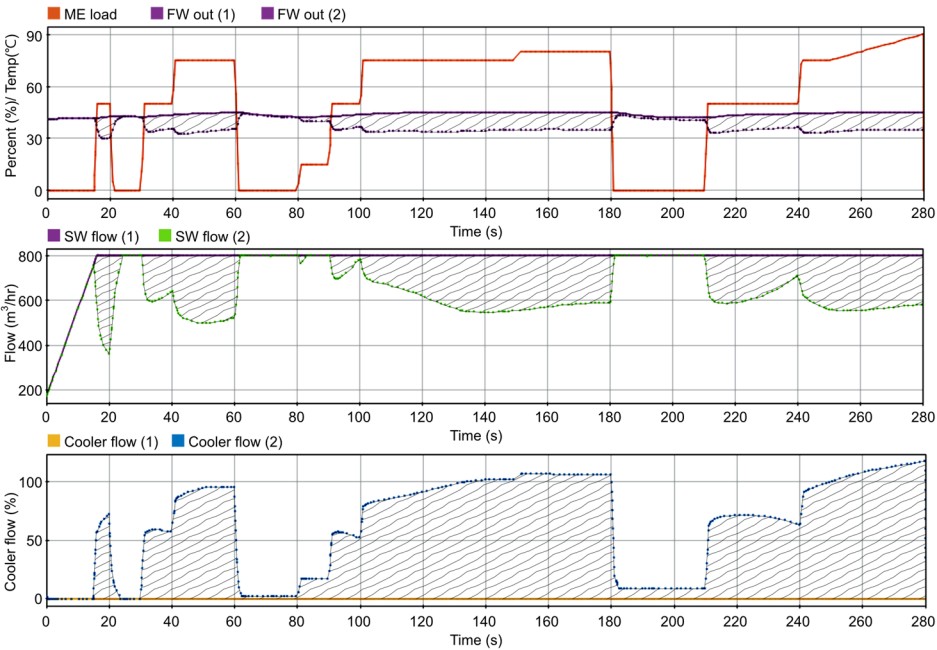

**Figure 11.** Experimental results of central cooling system with feed-forward controller.

Compared with the fresh water outlet temperature "FLOW (1)" of the existing system, the first graph shows that the dotted line of "FLOW (2)" of the system in which the feed-forward controller is installed approaches the desired value of 35 °C.

The second graph shows that the "SW flow (2)" is controlled and increases or decreases based on the load fluctuations of the main engine compared with the seawater pump flow rate "SW FLOW (1)" operated at 100% flow rate in the existing system.

In the third graph, compared to the cooler flow (1) of the existing system, it can be seen that "cooler flow (2)" reacts and opens in advance according to the load fluctuations of the main engine.

In summary, the opening degree of the three-way valve is controlled according to "ME load", and "cooler flow (2)" increases or decreases based on the load fluctuations of the main engine.

Figure 9 can be seen as a graph that occurs when only PID is installed, whereas Figure 11 can be seen as a graph when a feed-forward controller is installed. The top graph is the FW outlet temperature according to M/E and G/E load fluctuations, the solid blue line is the temperature graph when PID is applied, and the dotted blue line is the temperature controlled by about 22% lower than the solid blue line.

In other words, it was confirmed that when the feed-forward controller was applied, the outlet temperature of F.W was effectively controlled by about 22%. In addition, the middle graph shows the flow rate of the variable seawater pump. The purple solid line is the case when PID is applied, and the green dotted line is the graph showing the case when the feed-forward controller is applied.

When the variable speed seawater pump is operated with a green dotted line, energy savings of about 22% compared to the purple solid line can be seen. In addition, the bottom graph is the yellow line showing the cooler's three-way opening degree when PID is applied, and the blue dotted line is the graph with the feed-forward controller applied. In other words, the feed-forward controller has a faster adaptive control than the case of applying PID, and thus energy savings can be seen.

In other words, effective energy saving is achieved by controlling the seawater flow rate as the heat energy entering the central cooler increases or decreases.

## 5. Conclusions and Recommendations

Due to a disturbance called load fluctuation (main engine load), the output process affects both the three-way valve controller and the variable-speed seawater pump controller. The reason for the study of this paper was that the model ship Hannara applied the three-way valve controller and the variable-speed seawater pump controller with the PID control method, but the variable seawater pump fully continued to operate and the cooler's three-way valve was almost closed. Such behavior often have occurred on board ship. To overcome this phenomenon, effective results were obtained by applying a feed-forward controller according to the load fluctuation of M/E and G/E to both devices. When comparing PID control method in Figure 9 and the feed-forward control method in Figure 11 that puts the loads of M/E and G/E as input variables in advance. In this case, a variable seawater pump could save energy by about 22%.

As a result of the experiment by installing a feed-forward controller for disturbances so that the influence of these output variables is first reflected in the three-way valve controller, it was confirmed that the flow rate of the variable-speed seawater pump was smaller compared to that before the installation.

By applying a feed-forward controller, the sensitivity of the three-way valve controller to disturbances is increased, reducing the energy consumption of the seawater pump and solving the interference phenomenon between the controllers.

However, additional disturbances, such as a decrease in efficiency due to an increase in the pollution degree of the seawater side of the central cooling water cooler or a change in seawater temperature, may affect the performance of the feed-forward controller.

**Author Contributions:** Conceptualization, B.-G.J. and C.-M.L.; methodology, Y.-C.L.; software, T.-Y.J.; validation, C.-M.L., T.-Y.J.; formal analysis, T.-Y.J.; investigation, C.-M.L.; resources, C.-M.L.; data curation, Y.-C.L.; writing—original draft preparation, C.-M.L.; writing—review and editing, C.-M.L.;

visualization, C.-M.L.; supervision, Y.-C.L.; project administration, Y.-C.L.; funding acquisition, Y.-C.L. All authors have read and agreed to the published version of the manuscript.

**Funding:** This research received no external funding.

**Institutional Review Board Statement:** Not applicable.

**Informed Consent Statement:** Informed consent was obtained from all subjects involved in the study.

**Data Availability Statement:** This article was written by reconstructing the dissertation for master's degree by the first author, Chang-min Lee.

**Conflicts of Interest:** The authors declare no conflict of interest.

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
