# Peer review of "Design of Energy Saving Controllers for Central Cooling Water Systems"

_jmse, doi:10.3390/jmse9050513_

Round 1

Reviewer 1 Report

1, The introduction needs to be extended.

2, It is necessary to add a more detailed comment to listed curves at fig. 10, fig. 18 and fig. 20.

3, Schemes from the simulink are unnecessarily too large and confusing. For the reader, it would be appropriate to complete the overall principled block diagram of the regulatory process together with a commentary.

4, The conclusions only describe general conclusions from the research, while there is no discussion of the above statements, e.g. how much the energy consumption of the pump will be reduced or what effect the specified types of faults have on the performance!
The statements made in the conclusions should be confirmed by some analysis that would prove their validity, but this is missing from the article!

Author Response

please refer to attached file for responding reviewer 1

Reviewer 2 Report

It is necessary to elaborate Figure 7 in a more structural approach using flowcharts in connection with the Simulink models shown in Figure 8 and 9. Figure 14 is well done and/or such flowcharts will be added value to the contexts of the manuscript.

Some numerical highlights would be necessary. 

Reviewer 3 Report

In the reviewer's opinion, the article is interesting but has a design-engineering character with limited application of advanced theory.In the article authors have correctly modeled the cooling system with a variable load of the main engine and generators. Simulation experiments confirm the effectiveness of the proposed solution. However, the reviewer does not fully agree with the simulations.  Because in operational conditions on the ship it is not possible to change the engine load from 0% to 80% within 20 seconds. Also important factor that should be taken into account is the temperature of sea water feeding the cooler. According to the reviewer, the number of diagrams presenting the individual elements of the control system modeled in Matlab should be reduced. However, the theoretical section should be expanded. The  variables designation found in the text do not correspond with the  variables designation in the equations. This needs to be cleaned up.

Round 2

Reviewer 1 Report

The paper is acceptable in present form

Reviewer 3 Report

The authors correctly responded to the reviewer's comments, especially the added and expanded theory section. The indicated errors have been corrected. In this form the article can be published.